# Trim Flap System Design for Improving Ballistic-Lifting Entry Performance of the Tianwen-1 Mars Probe

**Xinli Li [1], Yansong Li [2], Dayong Hu [2,3,*], Wei Rao [1,*], Yufeng Qi [1], Qiang Yang [1] and Gang Wang [1]**

[1]  Beijing Institute of Spacecraft System Engineering, Beijing 100094, China; lxlsyl@126.com (X.L.); ggxxzz@vip.163.com (Y.Q.); yangq_cast@163.com (Q.Y.); wgtaurus@163.com (G.W.)

[2]  Department of Aircraft Airworthiness Engineering, School of Transportation Science and Engineering, Beihang University, Beijing 100191, China; lyswork@buaa.edu.cn

[3]  Aircraft/Engine Integrated System Safety Beijing Key Laboratory, Beijing 100191, China

[*]  Correspondence: hudayong@buaa.edu.cn (D.H.); rauwei@163.com (W.R.)

**Abstract:** The trim flap is an aerodynamic control surface capable of allowing Mars entry vehicles to improve landing performance during the most dangerous entry, descent, and landing (EDL) phase. In present work, a deployable trim flap system was proposed to meet the aerodynamical trim requirement of the Tianwen-1 Mars probe, which provided a mass-saving alternative to the conventional use of the center-of-gravity (CG) offset of ballast mass. In order to guide the trim flap design, theoretical and finite element (FE) models were established to predict and evaluate the deployment performance. Then, a full-scale physical prototype was manufactured for deployment experiments to verify the design effectiveness as well as validate the theoretical and FE models. Results predicted by theoretical and FE models were in good agreement with deployment experiments. Furthermore, the effects of three factors on the deployment performance were investigated, including the non-linear behavior of the damping, acceleration environment, and backshell flexibility. The manufactured prototype was installed on the Tianwen-1 Mars probe, saving more than 300 kg when compared to the conventional use of ballast mass CG offset, and assisted Tianwen-1 in achieving a successful landing, making China the first country in the world to utilize the trim flap technology for Mars EDL.

**Keywords:** trim flap; driving mechanism; deployment; Mars entry; Mars probe; entry; descent and landing

## 1. Introduction

As the closest planet to Earth in the solar system, Mars' environment is very similar to Earth [1]. Mars exploration is of great importance for understanding the origin of life and evolution, the discovery of extraterrestrial life, Mars colonization, and the mining of rare mineral resources. Consequently, Mars exploration has been the focus of planetary exploration for decades, representing the pinnacle of interplanetary space exploration and technology of the major aerospace countries [2]. Since the first Mars probe was launched by the Soviet Union in the 1960s, a total of 47 Mars probes have been launched by eight countries, 18 of which were aimed at safety landing on Mars, but only nine probes successfully landed on the surface of Mars (eight belonged to the United States and one belonged to China) [1], reaching a success rate of just 50%. China's first Mars exploration, using the Tianwen-1 Mars probe, was successfully launched from Wenchang Satellite Launch Center on 23 July 2020. When compared to previous Mars exploration missions, Tianwen-1 was the largest and most complex probe launched to Mars with a total weight of 5 tons, consisting of a 3.1-ton orbiter, a 1.6-ton lander, and a 240 kg rover. On 15 May 2021, the Zhurong Mars rover had safely landed at the expected landing site of Utopia Planitia on Mars, fulfilling the goals of "orbiting, landing and roving" in one mission. This marked China taking the first step toward independent planetary exploration. However, it should

be noted that there were still significant technical challenges for Mars entry vehicles during the EDL phase [2]. EDL has been considered to be one of the most dangerous and important phases for Mars exploration and directly determines the success of the entire exploration mission [1,3]. Reynier [4] and Salotti [5] summarized some Mars exploration missions and have revealed that many missions failed in the EDL phase. The primary reason was attributed to the relatively thin Mars atmosphere (approximately 1% of the density of Earth atmosphere), which resulted in the Mars entry vehicle decelerating at much lower altitudes with insufficient time and space to achieve safely landing on Mars [3,6,7]. In recent years, with the demand for robotic exploration missions, Mars sample returns, and even future human Mars missions, there was an urgent need to develop new EDL technologies for improving landing performance.

Figure 1 depicts the schematic of the EDL phases for different entry modes. For Mars entry vehicles, the current Mars entry strategies could be broadly classified into two categories: ballistic entry and ballistic-lifting entry. In ballistic entry, as shown in Figure 1a, entry vehicles entered at a 0° angle of attack to increase aerodynamic drag, so that no aerodynamic lift was generated to exercise any aerodynamic control over the flight path, resulting in a large landing dispersion in the order of several hundred kilometers [8]. Most previous Mars exploration missions, such as Pathfinder [9], Phoenix, and Mars Polar Lander adopted ballistic entry and achieved successful landing, demonstrating a good robustness. However, this was no longer suitable for future human landing requirements due to its landing accuracy limitation. Unlike the ballistic entry, the ballistic-lifting entry, adopted by the Viking landers (Viking I and Viking II), Mars Science Laboratory (MSL), and Mars 2020, could produce aerodynamic lift by flying with a non-zero angle of attack, and then the aerodynamic lift could be utilized to adjust the entry trajectory by changing the direction of lift vector, resulting in an order of magnitude improvement in landing accuracy over the Pathfinder and Phoenix ballistic entries [7,10]. The non-zero angle of attack was called a trim angle of attack, which allowed entry vehicles to trim aerodynamically and keep a steady flight state prior to parachute deployment. Usually, this trim angle of attack could be achieved by a radial offset of CG from the symmetry axis of the entry vehicle. For example, the Viking landers employed a combination of the stowed lander position and a configuration within the backshell and ballast mass to produce radial CG offset and obtain an expected trim angle of attack of 11°, as well as a lift-to-drag ratio (L/D) of 0.18 [11]. For MSL (see Figure 1b), CG offset was firstly created by jettisoning externally mounted cruise balance masses to maintain a desired trim angle of attack near to 16° after the cruise stage separation, which ultimately zeroed out by the sequential jettison of six tungsten masses just before parachute deployment [12]. The use of ballast mass to produce CG offset for aerodynamic trim, while mechanically simple, had an obvious disadvantage of sacrificing payload mass. Taking MSL as an example, the ejected ballast masses of about 318 kg was nearly 1.9 times the mass of Mars rover of 174 kg [13] and exceeded 30% of the lander payload mass of 900 kg [14], which not only resulted in a large payload sacrifice and cost waste, but also brought a challenge for the design of the interior structure layout of entry vehicles [15]. Consequently, it was very necessary to develop alternative methods with comparably less mass to provide similar or improved aerodynamic trim performance.

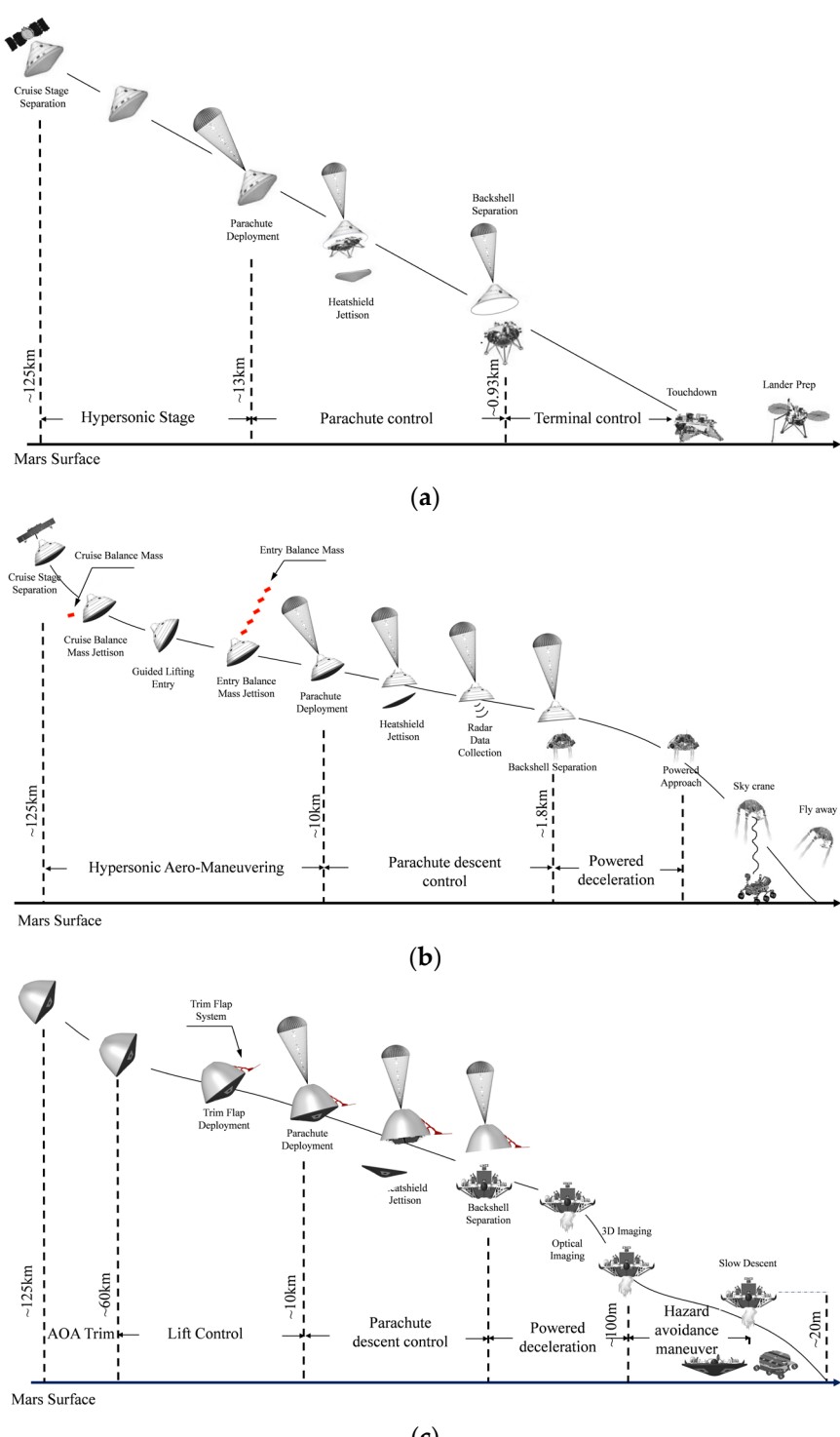

**Figure 1.** Sketch of three typical Mars entry, descent, and landing phases: (**a**) Ballistic entry; (**b**) Ballistic-lifting entry with the use of the center of gravity offset; (**c**) Ballistic-lifting entry with the use of the trim flap.

The trim flap is a deployable, aerodynamic control surface on the entry vehicle that can provide a direct control of the lift vector without requiring CG offset, allowing the entry vehicle to trim aerodynamically at a near constant angle of attack during EDL phase [11,13]. Figure 1c depicts a ballistic-lifting entry utilizing the trim flap, which had been considered in recent Mars missions and proposed for future missions [13]. One advantage was that the trim flap might eliminate the need for much of the ballast mass, allowing the mass savings to

be converted to usable payload. With a conservative mass estimate, the trim flap mass was on the order of tens of kilograms, as a comparation, the ballast mass consisted of hundreds of kilograms, indicating that the trim flap could significantly reduce payload sacrifice. In Mars Science Laboratory Improvements (MSL-I) project, NASA had conducted a conceptual design with a trim flap configuration that enabled a 1462 kg landing mass, as opposed to only 1086 kg for the baseline configuration with ballast mass CG offset. Intuitively, the use of the trim flap could obtain a payload mass gain of 376 kg. In addition, the trim flap system was about 30 kg, approximately one-tenth of the ballast mass, indicating the trim flap could reduce payload sacrifice up to 270 kg. This mass saving combined with the payload mass gain of 376 kg might be directly transformed into an actual payload mass increase of over 600 kg. When compared with the payload mass of one Mars rover of 174 kg, the mass saving and payload gain benefits for the trim flap were significantly greater than three scientific mission payload masses, and these mass savings could be directly transformed into payload mass. Further, another advantage was that the trim flap itself could produce aerodynamic drag due to the exposed flap area and subsequently aided in the deceleration of the entry vehicle. Based on above results, the trim flap had been believed as having potential for Mars missions, including scalability to larger class missions, as well as use at other solar system destinations [16].

Actually, the concept of the trim flap had been investigated as early as 1961 for Mercury and Apollo entry capsules [17]. In the Mars 2018 Project, Winski et al. [18] considered the use of a trim flap in place of CG offset to increase parachute deployment altitude and payload mass, as well as to reduce error in the EDL system. From their analysis results, payload mass could be increased by using a trim flap for entry masses below approximately 3230 kg in targeting the maximum parachute deployment altitude for a given entry mass and L/D. Murphy et al. [19] carried out wind tunnel tests and computational fluid dynamic predictions to examine the supersonic aerodynamic characteristics of the baseline Mar' 07 Smart Lander configuration with and without the fixed shelf/flap. Based on their results, it could be concluded that the flap appeared to be a feasible concept to meet the aerodynamic performance for mission requirements. Andersen and Whitmore [20] evaluated the effectiveness of the trim flap system and sized the flap using computational fluid dynamics and "first-order" engineering design tools based on incidence angle methods. Their result showed that a trim flap with the surface area of 1 $m^2$ could increase L/D from approximately 0.26 to 0.305. Similar work was also done by Horvath et al. [17] who performed aerodynamic wind-tunnel screening tests on a 0.029-scale model of the Mars Surveyor 2001 Precision Lander with a deployable flap to determine the effectiveness of the flap on the trim capability. Experimental results suggested a single flap could provide sufficient trim capability at the desired L/D for precision landing. However, this Mars exploration mission was cancelled. To further support supersonic aerodynamic database development for trim flap configurations, NASA Langley Research Center [11,14] carried out hypersonic wind tunnel tests on 38 unique trim tab configurations for scale blunt-body entry vehicles to parametrically evaluate the supersonic aerodynamic performance of trim flaps and to study the influence of flap area, aspect ratio, and cant angle. Force and moment data measured from wind tunnel tests indicated that the trim flap was a viable approach to improve the aerodynamic performance of blunt-body entry vehicles, and flap cant angle and flap area were found to be the most significant parameters affecting trim flap performance. The effect of flap configuration including the flap number, flap area, flap shape, and flap position on the trim aerodynamic characteristics were also investigated by Engel et al. [21] by using the Modified Newtonian aerodynamics code for an MSL-like vehicle. Results showed that the aerodynamic capability could be increased by using a high number of flaps, and by using flaps with large areas and moment arms. Recently, as Mars exploration missions became more complex, demands for large mass entry vehicles became stronger. Many novel entry vehicle configurations combined with the trim flap and other new EDL technologies were also proposed by NASA [10,22,23]. These studies confirmed the feasibility of trim flaps for Mars exploration missions and future human scale missions through computational aero-

dynamic analysis [18–21], wind tunnel experiments [11,14,17,19], and structural concept design [10,16,17,19,22,23].

However, to date, the trim flap has not been implemented in previous Mars missions and few concerns have been paid to the detailed structural design of the trim flap. To fill this gap in the research and to address the aerodynamic trim performance requirements of Tianwen-1 Mars probe, a deployable trim flap system was designed based on a single crank–rocker mechanism. Subsequently, the theoretical model based on the four-bar mechanism principle and FE model were established to analyze the deployment process of the trim flap system and to provide a guidance for the structure design, respectively. Then, to verify the effectiveness of the deployable trim flap system as well as to validate the accuracy of the theoretical and FE models, a full-scale physical prototype was manufactured and deployment experiments were also carried out. The experimental results were in good agreement with the analysis results. Furthermore, the effects of damping nonlinearity behavior, acceleration environment, and backshell flexibility on the deployment performance were also investigated. The physical prototype of the trim flap system in the present work was finally adopted in the Tianwen-1 Mars mission, and its successful landing proved the effectiveness of the trim flap system. To date, the Tianwen-1 Mars probe was the only probe in the world that used the trim flap technology for EDL. The present work built a solid basis for the use of the trim flap system in other planet exploration missions.

## 2. Structure Design of the Trim Flap System

In this section, within the scope of this research, a deployable trim flap system was designed for allowing the Tianwen-1 Mars probe to trim aerodynamically at a desired angle of attack during the EDL phase.

### 2.1. Design Requirement

The trim flap system would be designed to articulate from a stowed configuration to a deployed configuration. It was stowed in the backshell during launch and was not activated to exo-atmospherically deploy until the EDL phase, while it remained deployed during entry, as shown in Figure 1c.

The following were the basic design requirements for the trim flap system during the EDL phase:

1. One of the most fundamental design requirements was being able to deploy within 1.5 s. A time of 1.5 s was chosen as the deployment time (*Dt*) in order to minimize aerodynamic disturbances to Mars entry vehicles;
2. The flap panel deployed quickly in the Martian atmosphere at the hypersonic condition, at which time, the prevailing dynamic pressure exerted a counter force on the flap panel, preventing it from deploying. The driving force should be large enough to overcome this resistance and deploy the flap panel to its fully deployed position within 1.5 s;
3. A high-speed impact between the flap panel and the backshell as a result of a high-speed deployment should be eliminated [16];
4. Once fully deployed, the flap panel should remain in the full deployed configuration to withstand entry aerodynamic loads;
5. Other considerations included thermal protection, strength and stiffness, non-geometric interference with backshell and other components, etc.

### 2.2. Structure Description
#### 2.2.1. Schematic Diagram

To meet the design requirements, a deployable configuration, usually used in space structures [24–27], was proposed for the trim flap system based on a single crank–rocker mechanism. The schematic diagrams of the stowed configuration and the fully deployed configuration are shown in Figures 2a and 2b, respectively. As seen from Figure 2a, the whole trim flap system could be divided into two parts: a four-bar mechanism and a

driving mechanism. The four-bar mechanism was actuated by the driving mechanism to drive the flap rotational motion to fully deployed position, at which the dead point of the four-bar mechanism was reached and the four-bar mechanism was locked, forming a "triangular" configuration to withstand severe aerodynamic load.

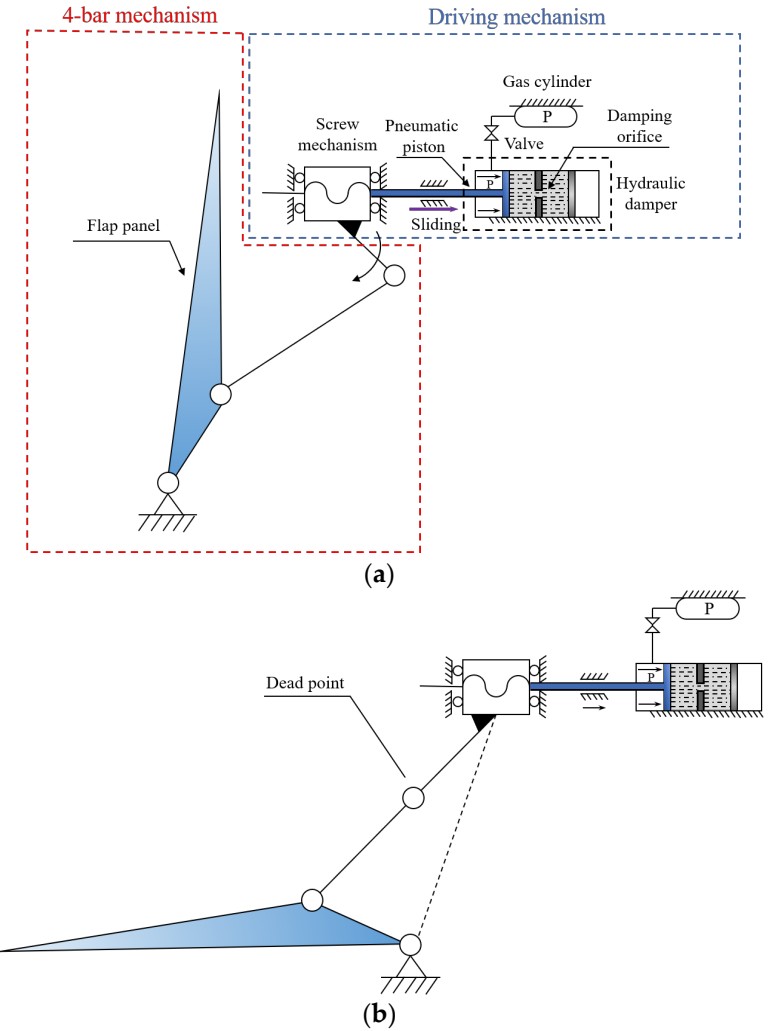

**Figure 2.** The schematic diagram of the trim flap system: (**a**) Stowed configuration; (**b**) Fully deployed configuration.

### 2.2.2. Four-Bar Mechanism

The geometry representation of the fully deployed configuration, stowed configuration, and the corresponding exploded view of the trim flap system are presented in Figure 3a–d, respectively. As illustrated in Figure 3d, the four-bar mechanism was mainly composed of the following components: a crank, connecting rod, curved beams, flap panel, rib plate, and supports. The driving mechanism and supports (see 1 and 6 in Figure 3b) were bolted to the backshell, and the flap panel, as shown in Figure 3e, was fastened on the backshell through the electric blasting valve (see 7 in Figure 3b), which maintained the trim flap system in a stowing configuration before deployment. The crank and connecting rod, curved beams and supports, rib plate, and connecting rod were all linked by hinges (see B, C, and D in Figure 3b), while the flap panel was connected with curved beams and rib plate through bolts, respectively. This resulted in three revolute joints, forming a four-bar mechanism. One end of the crank was attached to the driving mechanism (see Point A in Figure 3b), and the driving mechanism could produce actuation force to drive the crank to rotate around Point A.

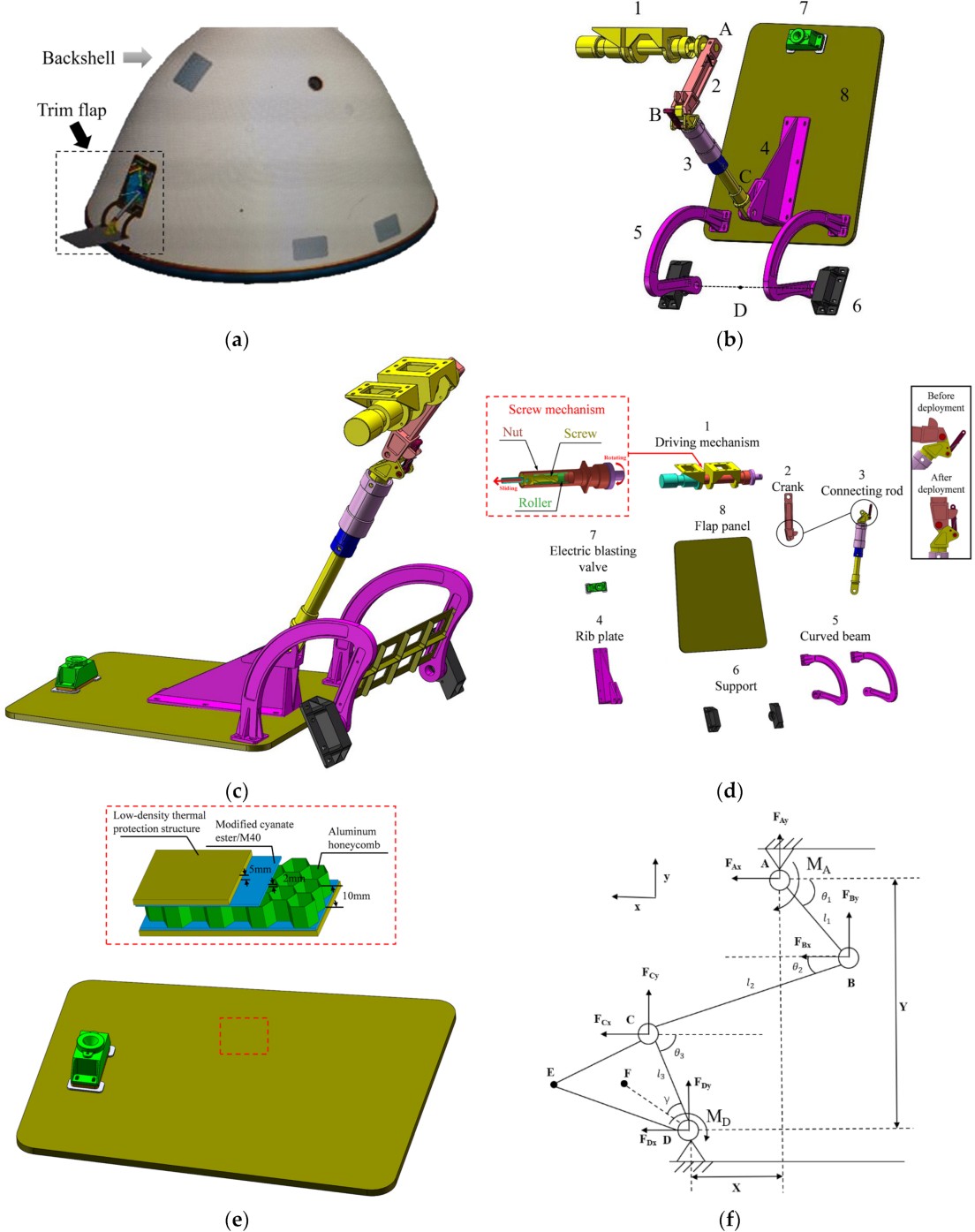

**Figure 3.** Geometry and theoretical model of the trim flap system: (**a**) Fully deployed configuration of the trim flap system mounted to the backshell; (**b**) Stowed configuration of the trim flap system; (**c**) Deployed status of the trim flap system; (**d**) Exploded view; (**e**) Detailed sandwich structure of flap panel; (**f**) Simplified planar two-dimensional theoretical model.

### 2.2.3. Driving Mechanism

The driving mechanism, as the actuation system, is the key component of the whole trim flap system. It should provide enough driving force to rotate the flap panel relatively quickly for achieving a required angle of attack, as well as to resist hypersonic aerodynamic forces on the flap panel [28]. Additionally, it should act as a damper to generate enough viscous resistance to eliminate the injury to the flap panel and backshell induced by impact

forces due to high deployment velocity, while still meeting the design requirement of *Dt* within 1.5 s.

The schematic diagram of the driving mechanism is shown in Figure 2a. The driving mechanism consisted of a pneumatic piston, hydraulic damper, screw mechanism, and high-pressure gas cylinder. The pneumatic piston was adopted as an actuator to provide driving force, because it had some mechanical advantages, such as installation flexibility, fast *Dt*, and a reasonable mass, as compared with other actuation methods, such as ball screws and hinge actuators [16]. The piston was pushed by high pressure air stored in the gas cylinder to make a linear motion along its axial axis. Then the screw mechanism converted the linear motion to rotational motion, and a linear force to a torque, with the roller linked to the piston and the nut attached to the crank, as highlighted in Figure 3d. The relationship between the air pressure on the piston and the torque that drove the crank rotation could be expressed as:

$$M_0 = PA\left(\frac{\tau}{2\pi}\right) \tag{1}$$

where $M_0$, $P$, and $A$ represent the ideal driving torque acting on the crank, air pressure on the piston, and piston area, respectively. $\tau$ was the pitch of the screw pair with the value of 126.3 mm, which meant that in one period, the piston moved 126.3 mm, leading to the crank being rotated by an angle of $2\pi$. The relationship between the moving speed of piston and the angular velocity of crank could be expressed as:

$$\frac{2\pi}{\omega} = \frac{\tau}{V} \tag{2}$$

where $V$ is the linear motion speed of piston and $\omega$ is the angular velocity of the crank.

The hydraulic damper connected to the piston was used to limit excessive deployment velocity while ensuring that *Dt* could meet the requirement. The damping resistance was generated through acting on the piston when the piston compressed the oil chamber and squeezed the silicone oil through damping orifices, as shown in Figure 2a. Similar to Equation (1), the damping resistance provided by the hydraulic damper could also be converted to damping torque $M_c$ acting on the crank through the screw mechanism:

$$M_c = C_t V \cdot \frac{V}{\omega} = C_t \left(\frac{V}{\omega}\right)^2 \omega = C_t \left(\frac{\tau}{2\pi}\right)^2 \omega \tag{3}$$

where $C_t$ is the damping coefficient of the hydraulic damper. $C_t$ is affected by many factors, such as the size of damping orifice, the density and temperature of silicone oil, and the viscosity and shear rate of silicone oil. It usually exhibits a complex nonlinear behavior [29]. For simplicity, a constant $C_t$ was adopted in the theoretical analysis during the preliminary design stage.

From Equations (1) and (3), the actual driving torque $M_A$ applied to the crank could be obtained from the subtraction of ideal driving torque from the damping torque, multiplied by the transmission efficiency:

$$M_A = (M_0 - M_c)\eta = [PA\left(\frac{\tau}{2\pi}\right) - C_t \left(\frac{\tau}{2\pi}\right)^2 \omega]\eta \tag{4}$$

where $\eta$ is the transmission efficiency and represents the ratio of the ideal driving torque to the actual driving torque [30,31]. The value of $\eta$ was set as 80% to consider the efficiency loss of the driving mechanism. From Equation (4), clearly, the faster $\omega$ is, the higher $M_c$ is generated, leading to lower $M_A$. Thus, excessive deployment velocity could be limited, reducing the impact force to the backshell and flap panel.

### 2.2.4. Deployment Process

The deployment process of the trim flap system was as follows:

1.  The deployment was initiated exo-atmospherically with the firing of the electric blasting valve that released the preloaded launch lock interface;

2. This was immediately followed by the valve of the gas cylinder being opened, the gas flowing into the pneumatic piston, and the air pressure creating an actuating force on the piston to drive the crank rotation through the screw mechanism;

3. After the crank was actuated, based on the characteristic of the crank–rocker mechanism, the flap panel was driven to deploy to its fully deployed position;

4. When the flap panel reached the position perpendicular to the symmetry axis of the backshell (as shown in Figure 3a), corresponding to the dead point of the crank–rocker mechanism, the flap panel would be locked with a self-lock mechanism as highlighted in Figure 3d and remained in its fully deployed status to withstand aerodynamic load during entry.

### 2.2.5. Material Selection

All the components were made of aluminum alloy with the density of 2700 kg/m$^2$ and elastic modulus of 72,000 MPa listed in Table 1, except the flap panel. The flap panel was made of a sandwich structure with an aluminum honeycomb core and modified cyanate ester/M40 facets due to high specific strength and stiffness, while a low-density thermal protection structure was used on the exposed panel surface to protect the flap panel against high aerodynamic heat expected at the hypersonic speed. The mechanical properties of the modified cyanate ester/M40 and honeycomb core panel are also listed in Table 1. The total mass of the whole trim flap system was 10 kg, which was only about one-quarter of that of the MSL-I designed by NASA [16]. And, the weight saving was more than 300 kg, as compared to the conventional use of ballast mass CG offset (MSL ejected a ballast mass of about 318 kg [12]). Consequently, the Tianwen-1 Mars probe, utilizing this trim flap system, provided the most payload mass due to the mass savings over replacement of the entry balance masses.

**Table 1.** Material mechanical properties.

| | $\rho$ kg/m$^3$ | $E_1$ MPa | $E_2$ MPa | $G_{12}$ MPa | $G_{13}$ MPa | $G_{23}$ MPa | $\mu$ |
|---|---|---|---|---|---|---|---|
| Modified cyanate ester/M40 | 550 | 230,000 | 7000 | 4000 | | | 0.3 |
| Epoxy resin/M55 | 1640 | 240,000 | 7000 | 4600 | 3833.33 | 3833.33 | 0.3 |
| Aluminum honeycomb | 27 | 0.0001 | 0.0001 | 0.0001 | 140 | 76 | 0.3 |
| Aluminum alloy | 2700 | 72,000 | | | | | 0.3 |

## 3. Theoretical Model and FE Model

In this section, a theoretical model and FE model have been developed to analyze the motion of the trim flap and to provide a guidance for a detailed structure design.

### 3.1. Theoretical Model

#### 3.1.1. Kinematic and Dynamic Analysis

The motion mechanism of the trim flap system was essentially a four-bar mechanism which is similar to the trailing-edge flap system of airplanes [30]. Therefore, a planar two-dimensional model as shown in Figure 3f was established for kinematic and dynamic analysis using the Lagrange method. Bar AB and BC represented the crank and connecting rod, respectively. Part CDE represented the combination of the flap panel, rib plate, and curved beams. Because these components moved together and the contribution of the flap panel to the motion mechanism mainly came from its mass, the following analysis only involved the mass of the flap panel without considering the influence of its shape, and the motion of part CDE would be represented by that of rod CD. The revolute joints at points A, B, C, and D were all set as hinges, while the translation degrees of points A and D were fixed, thus forming a four-bar mechanism. A coordinate system is established in Figure 3f to describe the position of each part of the mechanism. Here $l_1$, $l_2$, and $l_3$ represent the length of AB, BC, and CD, respectively; $\theta_1$, $\theta_2$, and $\theta_3$ are the angles between AB, BC, CD, and the $X$-direction, respectively. $\dot{\theta}_1$, $\dot{\theta}_2$, and $\dot{\theta}_3$ represent the angular velocity of AB,

BC, and CDE, respectively, while $\ddot{\theta}_1$, $\ddot{\theta}_2$, and $\ddot{\theta}_3$ represent the angular acceleration of AB, BC, and CDE, respectively. $\Delta\theta_3$ represents the deployment angle of the flap panel and is equal to the increment of angle $\theta_3$ relative to its initial value. $X$ and $Y$ are the distance between point A and D in the $X$-direction and $Y$-direction, respectively. $F_{Ax}$, $F_{Ay}$, $F_{Dx}$, and $F_{Dy}$ represent constraint reaction forces from the driving mechanism and supports at hinge A and D, respectively. $F_{Bx}$, $F_{By}$, $F_{Cx}$, and $F_{Cy}$ are internal forces that act at hinge B and C. $g$ is the acceleration environment and its value was 9.8 m/s$^2$ on the surface of the earth and 3.7 m/s$^2$~19.6 m/s$^2$ during the aerodynamic deceleration stage under Martian atmospheric conditions. In addition to $M_A$ at Point A, an external moment $M_D$ to resist the flap panel deployment was applied at hinge D to simulate the aerodynamic resistance exerted on the flap panel and an approximately linear relationship between the aerodynamic resistance and $\Delta\theta_3$ was obtained from the aerodynamic experimental data:

$$M_D = \alpha \cdot \Delta\theta_3 \tag{5}$$

where $\alpha$ was a constant. With the consideration of the uncertainties of Martian atmosphere [32], the values of $\alpha$ were 51.6 and 9 N·m/rad for maximum and minimum aerodynamic resistance condition, respectively.

The dynamic equations of motion could be established using the Lagrange method and solved by the classical fourth-order Runge-Kutta method using MATLAB in Appendix A. The results were also validated by SolidWorks motion simulation to avoid geometric interferences with the backshell and other components.

3.1.2. Parametric Analysis

$P$ and $C_t$ provided by the driving mechanism were two key parameters affecting the structure responses. In terms of $P$, it not only enables the flap panel to deploy quickly within the allowed time, but also can resist the aerodynamic forces acting on the flap panel that prevent it from deploying [28]. It should have a sufficient margin for two reasons: one was the effect of the complexity and high uncertainty of Mars mission environments [32], which was mainly due to our weak technical background and lack of first-hand experience and data in these areas [2]; the other was attributed to the pressure drop caused by gas leakage and low temperature. Here, the maximum pressure of 30 MPa was selected to ensure a high deployment velocity and a short $Dt$ for the whole trim flap system.

However, the high deployment velocity inevitably induced a large impact on the structures of flap panel and backshell, leading to structural damage and failure, e.g., large deformation or even fracture of the flap panel, especially bolts pulled out from the sandwich composite of the backshell when the constraint reaction force $F_A$ at bolt connections exceeded 1500 N. Accordingly, in addition to the $Dt$ requirement, the limit of the constraint reaction force within 1500 N was considered as another design constraint. For this reason, viscous resistance was required to decelerate the deployment velocity and to eliminate the structure, injury while ensuring $Dt$ within 1.5 s.

A parametric analysis was carried out to investigate the effect of $C_t$ on the deployment performance, in order to provide a reference for selecting appropriate $C_t$. Two extreme external loading conditions were adopted to account for the pressure drop of $P$ caused by a low temperature and gas leakage, as well as the influence of uncertainties of the Martian atmosphere on aerodynamic resistance, as follows:

(1)　Minimum $Dt$ condition: external loads were a combination of $P$ of 30 MPa and a minimum $\alpha$ of 9 N·m/rad, corresponding to the fastest deployment process.

(2)　Maximum $Dt$ condition: Considering the harsh environment in Mars exploration missions, $P$ might drop from 30 to 15 MPa due to an extremely low temperature and gas leakage. Under the maximum $Dt$ condition, $P$ of 15 MPa and maximum $\alpha$ of 51.6 N·m/rad were used for the external loads, corresponding to the slowest deployment process.

Figure 4 presents the time history curves of $\Delta\theta_3$ with different $C_t$ under the minimum *Dt* condition. It could be observed that $\Delta\theta_3$ of the flap panel increased from 0 and finally reached 1.99 rad when fully deployed. *Dt* showed an increasing trend with the increase of $C_t$, demonstrating the slowdown effect of damping. When reaching the same $\Delta\theta_3$ of 1.99 rad, *Dt* increased from 0.32 to 0.85 s with the increase of $C_t$ from $1.0 \times 10^5$ to $3.0 \times 10^5$ N·s/m. From the slope of each curve in the figure, it could be intuitively observed that the deployment velocity showed a decreasing trend with the increase of $C_t$.

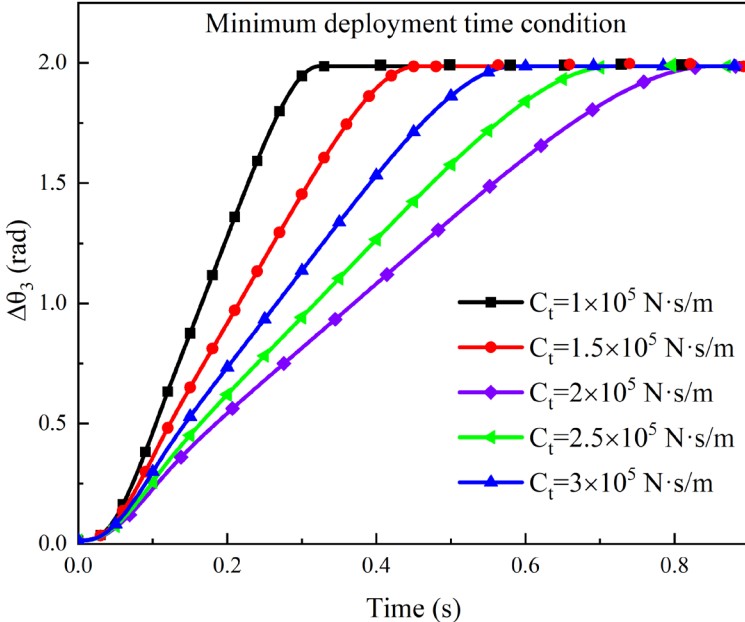

**Figure 4.** Time history curve of the deployment angle under different $C_t$ under minimum *Dt* condition.

Figure 5a,b showed the variation curves of $F_A$ and *Dt* with respect to $C_t$ under different loading conditions, respectively. As seen from the figures, as $C_t$ increases, *Dt* increases while $F_A$ decreases, indicating that the damping reduced the deployment velocity and the impact force on the backshell. Moreover, by comparing the results under maximum *Dt* condition and minimum *Dt* condition, it could be observed that *Dt* of the former was higher than that of the latter, but $F_A$ exhibited the opposite trend. In fact, the results of the actual loading condition should be found in the shade area between two extreme external loading conditions, as shown in Figure 5. To meet the requirements of *Dt* and $F_A$ of less than 1.5 s and 1500 N, respectively, it could be seen that the allowed $C_t$ were determined to be in the range from $1.25 \times 10^5$ N to $2.25 \times 10^5$ N·s/m and this was provided to the manufacturer as a reference for their detailed design of the hydraulic damper.

### 3.2. FE Model

As compared with the theoretical model, the FE method could consider more detailed structural deformation characteristics, stress distributions, and complex interaction behavior. In this section, an FE model with these design parameters was also developed by using the software ABAQUS to simulate the deployment process of the trim flap system, in order to verify the effectiveness of the proposed theoretical model and investigate the mechanical characteristics.

Figure 6 shows the FE model of the trim flap system, which was fixed on an aluminum alloy frame. It mainly consisted of a crank, connecting rod, curved beams, flap panel, rib plate, and supports. As a typical thin-walled structure, it was reasonable to adopt shell elements with a linear elastic constitutive model to set up the FE model, in order to drastically reduce the computation cost. Joints in the trim flap system were made of high-strength steel, which were strong enough not to fail during the deployment process,

compared to thin-walled structures such as the cranks and rockers. As a result, in the FE model, joints were simplified as the non-deformable revolute joint connectors. The revolute joint connector was a type of connector element that joined the position of two points and only allowed a relative rotation of the connection in the local coordinate *X*-direction. The driving mechanism was simplified as a rigid point (Point A) attached to the crank, because the driving mechanism was very complex, and its mechanical response was not an issue of concern in the present work. A revolute joint was employed to link Point A to the ground. $M_0$ provided by the driving mechanism was calculated according to Equation (1) and directly applied on Point A to drive the crank rotation, while $C_t$ was defined as the connector damping behavior of the revolute joint to simulate the damping behavior of the hydraulic damper. Similarly, three revolute joints were also defined, respectively, between the frame and curve beams (revolute joint D), the crank and connecting rod (revolute joint B), and the rib plate and connecting rod (revolute joint C). These three revolute joints correspond to the hinges of the geometry model (as shown in Figure 3b).

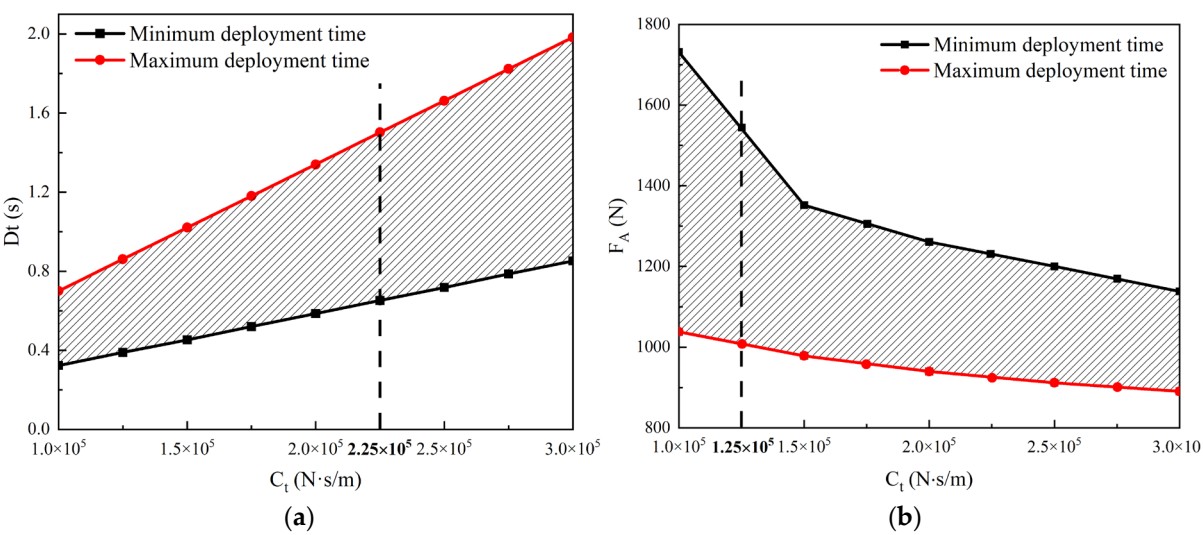

**Figure 5.** Parametric analysis of the effect of the $C_t$ on $Dt$ and $F_A$ under different external loading conditions: (**a**) $Dt$; (**b**) $F_A$.

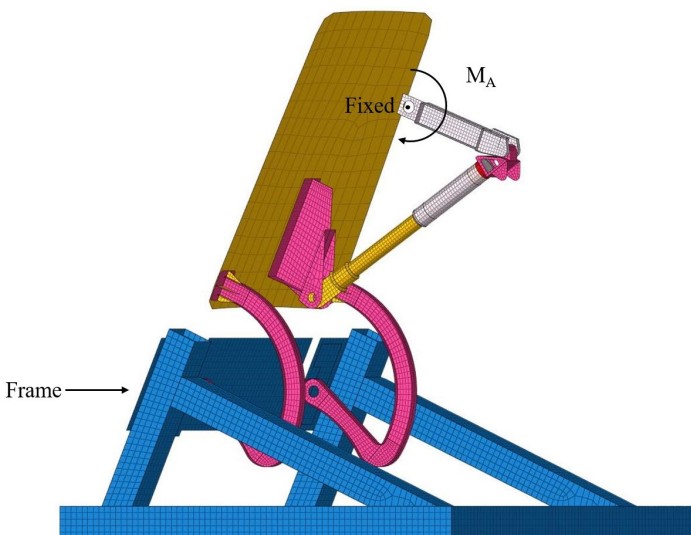

**Figure 6.** FE model of the trim flap system fixed on the aluminum alloy frame.

For the flap panel, four-node laminated shell elements were used to model the sandwich structure, where layers 1 to 4 and 6 to 9 were the modified cyanate ester/M40 with

the thickness of 0.08 mm per layer, and the aluminum honeycomb core was simplified as a homogenous material with the thickness of 9.36 mm. The mechanical properties used in the FE model were summarized in Table 1.

The aluminum alloy frame was fully clamped by restraining all degrees of freedom of the bottom nodes. To model the interaction of the self-locking mechanism, a penalty-based general contact was defined, including the crank and connecting rod. $M_D$, indicating the ability to resist the deployment of the flap panel could be simulated by defining a rotational stiffness equal to $\alpha$ for the rotational stiffness behavior of the revolute joint D, and thus $M_D$ could be obtained from the product of the rotational stiffness and the rotation angle (see Equation (5)). The constraint reaction force at Point A, the rotation angle, and the angular velocity of the revolute joint D were selected as output at a time interval of 0.001 s. ABAQUS/Standard with the automatic time step was chosen as the solver and a SAE 600 filter was also used to eliminate the numerical oscillation.

To validate the FE model, modal analysis was conducted under the fully deployed configuration and the stowed configuration, respectively, and compared with the corresponding modal tests. For the stowed configuration, the first order frequency of 58.7 Hz obtained from the FE model was very close to experimental result of 60 Hz, while for the fully deployed configuration, the first order frequency of 21.6 Hz obtained from the FE model also agreed well with experimental result of 21.9 Hz. The high agreement between the numerical analysis and experiments proved the effectiveness of the FE model and built confidence in the use of the FE model to simulate the deploy process for the trim flap system.

## 4. Results and Discussion

### 4.1. Deployment Experiment

Based on the theoretical and FE analysis, a physical prototype of the trim flap system proposed in Section 2 was manufactured and used for deployment experiments to verify the deployment performance. Figure 7 showed the fully deployed configuration of the trim flap system mounted to an aluminum alloy frame through bolt connections at the driving mechanism and supports. The minimal aerodynamic resistance $M_D$ (see Equation (5)) was adopted and simulated by a servo motor located at Point D (see Figure 3b) to generate the moment linearly related to the deployment angle of the flap panel. For the driving force, a maximum pressure of 30 MPa with fluctuations was adopted to consider the effects due to low temperatures and gas leakage. Mark points were pasted on the rib plate, flap panel, and curved beams and a high-speed camera with a sampling frequency of 1000 fps was used to capture the motion state. Then, a motion analysis system named TEMA was used to obtain kinematic data by tracking the position of the marked points. Three experiments were conducted to ensure repeatability.

Figure 8a presents the deployment process of the trim flap system captured by the high-speed camera. It could be seen from the Figure 8a that the flap panel actuated by the driving mechanism deployed gradually until it reached full deployed status and was then locked to the fully deployed configuration. No structure failure was found during the deployment process. It could be considered that the trim flap system could be successfully deployed under the current load condition.

Figure 9 presents the time history curves of $\Delta\theta_3$ and $\dot{\theta}_3$ processed from the mark points' data. Clearly, the results exhibited a good consistency, although there were still some slight differences. The differences were mainly caused by the clearances of the driving mechanism and hinges [26], while the error of air pressure also contributed to the differences. As seen in Figure 9, the maximum $\Delta\theta_3$ was about 1.99 rad, which was consistent with the theoretical analysis (See Figure 4). $Dt$ corresponded to the time taken to reach the maximum $\Delta\theta_3$, which was about 0.45 s in Figure 9, satisfying the basic design requirement for $Dt$ within 1.5 s. It could be also observed from Figure 9 that the curve of $\dot{\theta}_3$ was characterized by a fluctuation pattern with peaks and troughs. When the deployment started, as $M_c$ was relatively low due to the low velocity (see Equation (3)), $M_0$ was much higher than $M_c$,

resulting in a rapid increase in the angular velocity of the crank. As the angular velocity increased, $M_c$ kept increasing until it approached $M_0$. The peak of $\dot{\theta}_3$ occurred when $M_c$ was approximately equal to $M_0$ at 0.08 s. After that, based on the kinematic characteristics of the four-bar linkage mechanism, the deployment velocity varied continuously as the mechanism moved and the transmission angle changed. $\dot{\theta}_3$ finally dropped to zero when the crank was parallel to the connecting rod, corresponding to the position of the dead point.

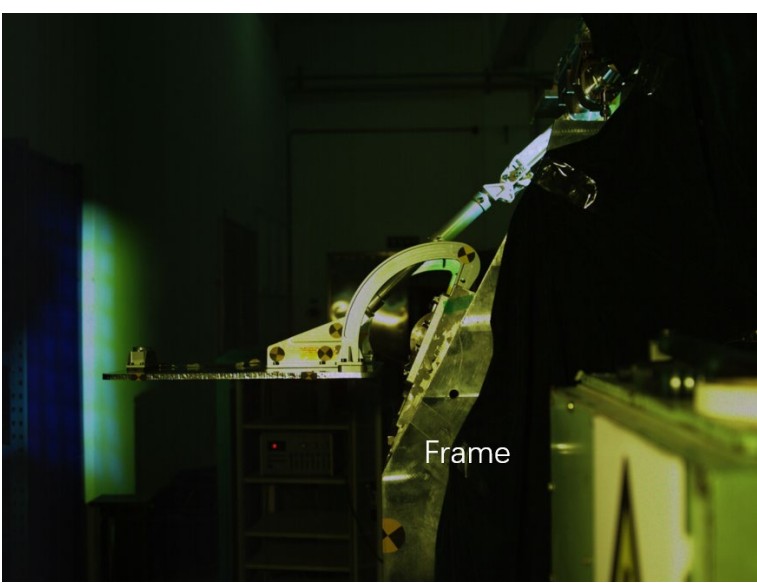

**Figure 7.** Physical prototype of the trim flap system.

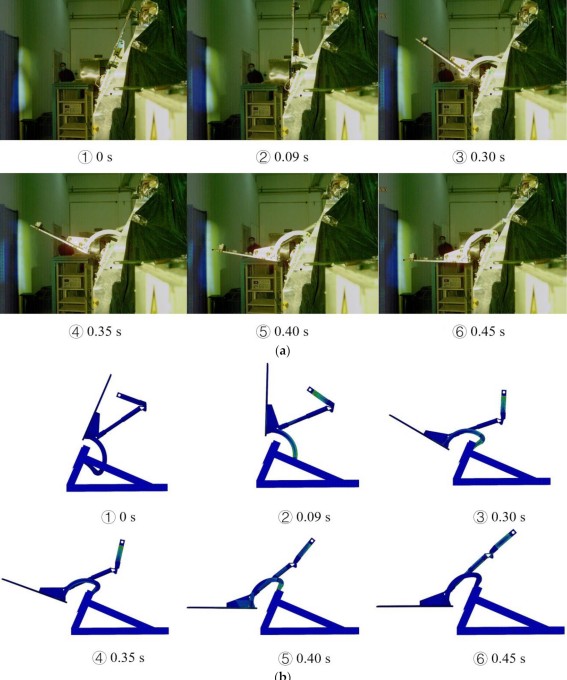

**Figure 8.** Comparison of deployment process of the trim flap system between experiment and numerical simulation: (**a**) Deployment process captured by high-speed camera; (**b**) Deployment process by numerical simulation.

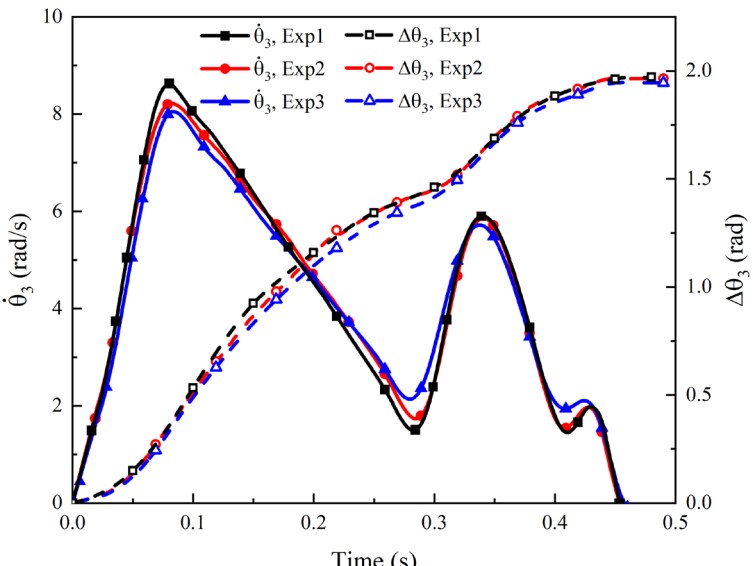

**Figure 9.** Time history curves of the deployment angle and angular velocity.

After the deployment experiments, the structure was carefully inspected and no obvious structure damage or failure were observed. It could be concluded that the designed prototype was effective and satisfied the design requirements under the current loading condition.

*4.2. Comparation between Analysis and Experiments*

The deployment experiment results could be used to validate the theoretical model and FE model. In the preliminary design phase of the structure, the constant $C_t$ was adopted in the theoretical analysis and the nonlinear behavior of damping was neglected for simplicity. However, in practice, the damping behavior of the hydraulic dampers was very complex. Therefore, to ensure that $C_t$ in the theoretical analysis and numerical simulation was the same as that in experiments, $C_t$ versus the crank angular velocity $\dot{\theta}_1$ was measured experimentally as shown in Figure 10. From the figure, it can be seen that $C_t$ exhibited an obvious nonlinear behavior. It was very large when $\dot{\theta}_1$ was lower than 0.5 rad/s, and gradually decreased with the increase of $\dot{\theta}_1$, tending to a stable value of $1.5 \times 10^5$ N·s/m after $\dot{\theta}_1$ exceeded 2 rad/s, which was within the recommended design range of $1.25 \times 10^5$ to $2.25 \times 10^5$ N·s/m. By fitting the curve using the least squares method, the relationship between $C_t$ and $\dot{\theta}_1$ was:

$$C_t = \frac{A}{\dot{\theta}_1} + B \tag{6}$$

where $A = 3.36 \times 10^3$, $B = 1.42 \times 10^5$. By substituting Equation (A33) into the FE model and theoretical model as well as taking the same setting as the experiments, the theoretical analysis and numerical simulation of the deployment process for the trim flap system were carried out and compared with the experimental results.

The comparisons of the results achieved from the experiment, as well as the theoretical and numerical simulations are presented in Figures 8 and 11, respectively. From Figure 8, obviously, the FE simulation could reproduce the global experimental phenomenon very well. The numerical prediction of the motion trajectory of the flap panel exhibited an excellently good agreement with experiments. Moreover, it can be found in Figure 11 that the curves of the deployment angle and angular velocity of the flap panel obtained from the theoretical analysis and FE simulation matched well with the experimental results qualitatively and quantitatively, except the results predicted by theory and numerical simulation were slightly higher than the experimental data. Based on the above results, it

could be concluded that the proposed theoretical model and FE model could be used as an effective, time-saving tool to provide a guidance for the design of the trim flap system, as well as to predict and evaluate its deployment characteristics.

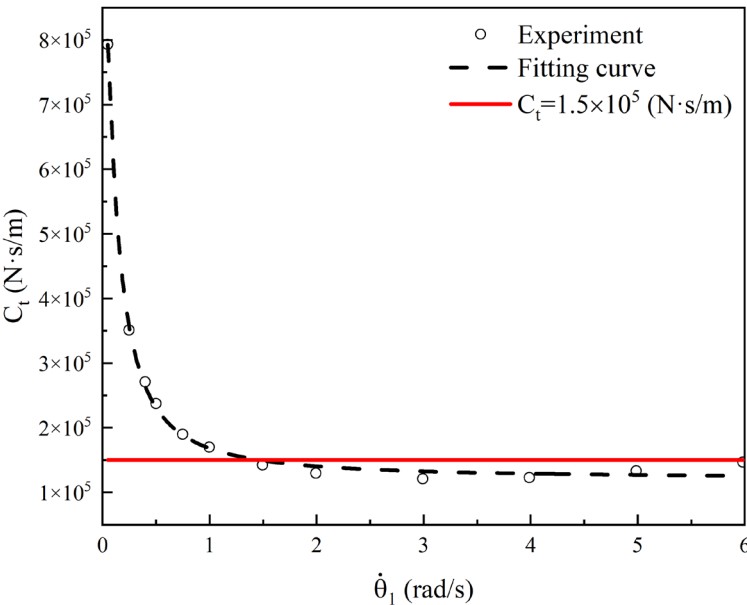

**Figure 10.** Damping coefficient vs. crank angular velocity for the driving mechanism.

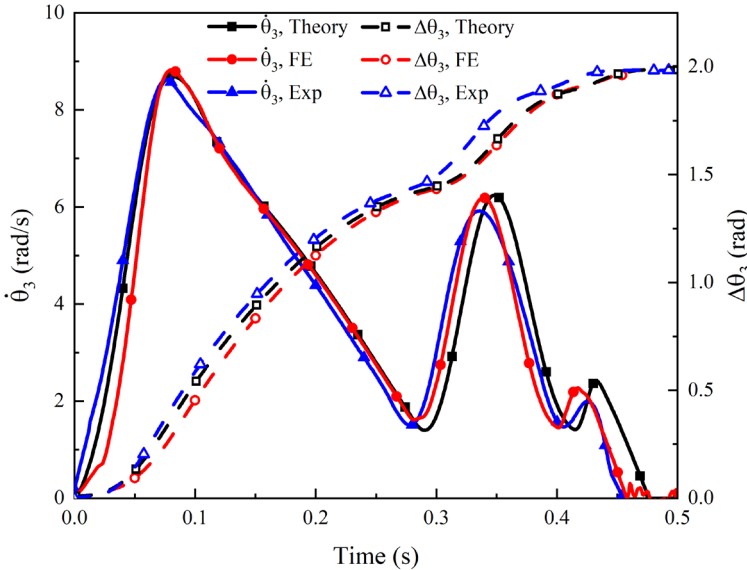

**Figure 11.** Comparison of the deployment angle and angular velocity of the flap panel from theoretical, FE, and experimental results.

Furthermore, the structure strength and stiffness of the trim flap system could also be predicted and evaluated by using FE results. The maximum stress located at the crank, the connecting rod, and the curved beams was 340.4, 219.6, and 37.4 MPa, respectively, all of which were within the limit of the material strength of 380 MPa. Additionally, no large structure deformation or failure was found in Figure 8b during the deployment process, similar to the experimental results. It could be considered that the structure strength of the trim flap system had a sufficient safety margin to allow the flap panel to deploy successfully without any damage or destruction during the deployment process.

### 4.3. Influence of the Nonlinear Behavior of Damping

The effect of the nonlinear behavior of damping on the deployment performance for the trim flap system was also investigated by comparing the results obtained from the theoretical model using constant $C_t$ of $1.5 \times 10^5$ N·s/m and nonlinear $C_t$ from Figure 10. The comparison of time history curves of the deployment angle and angular velocity of the flap panel for different $C_t$ are presented in Figure 12. It could be observed that the results calculated using constant $C_t$ were very close to those calculated using nonlinear $C_t$, and the maximum error between them was only 3.7%, demonstrating that the nonlinear behavior of damping at low $\dot{\theta}_1$ had relatively little effect on the deployment performance and could be neglected. Thus, from the viewpoint of saving test costs and time, only $C_t$ at high speeds needed to be measured and applied to predict and evaluate the deployment performance, which was also very convenient and effective for the preliminary design stage of the structure.

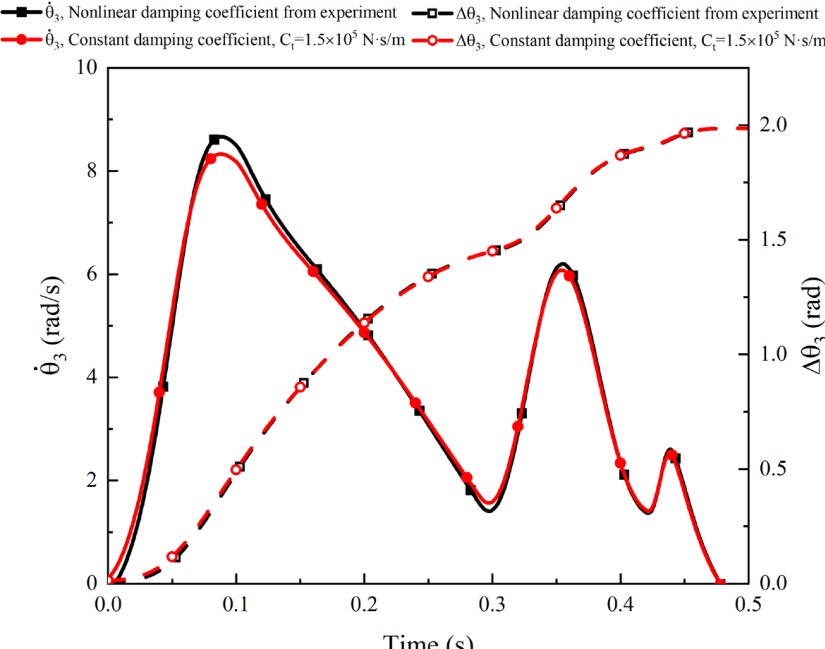

**Figure 12.** Comparison of the deployment angle and angular velocity between nonlinear $C_t$ and constant $C_t$.

### 4.4. Influence of Acceleration Environment

The deployment experiment of the trim flap system was carried out on Earth with a gravitational acceleration of 9.8 m/s², while during the EDL process, the trim flap was deployed under the aerodynamic deceleration of 3.7~19.6 m/s². It was necessary to examine the effect of different acceleration environments on the deployment performance of the trim flap system.

Figure 13 shows the time history curves of the deployment angle and angular velocity of the flap panel under three acceleration environments (3.7, 9.8, and 19.6 m/s²). Clearly, the calculated results under all three acceleration environments were very close and the maximum error was within 3.6%, which indicated that different acceleration environments had very little effect on the deployment performance. The reason was because the potential energy associated with the acceleration environment was less than 3% of the input work and thus could be ignored.

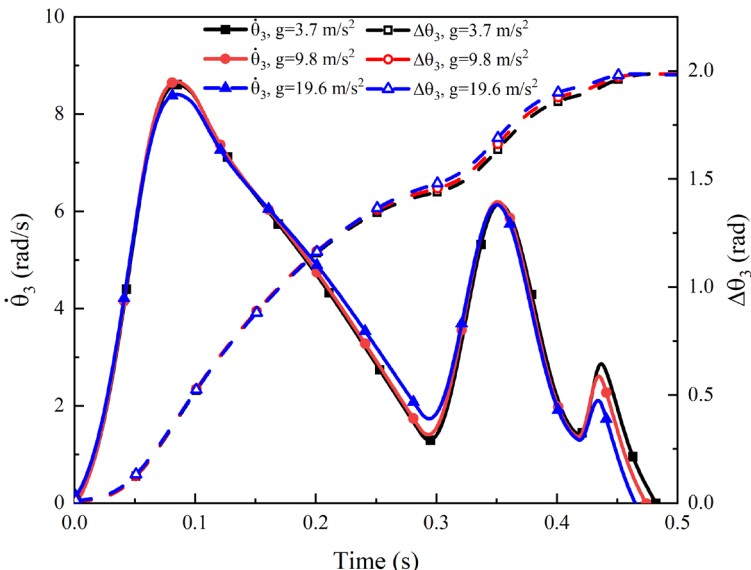

**Figure 13.** Comparison between different acceleration environments.

### 4.5. Influence of the Backshell Flexibility

In the deployment experiment, the trim flap system was fixed on an aluminum alloy frame, which could be considered as a rigid base due to its relatively great stiffness. However, in actual applications, the trim flap system was installed on the backshell of the Tianwen-1 Mars probe, and thus the backshell flexibility might have an influence on the deployment performance.

To address this issue, a comprehensive full-scale 3D FE model including the backshell and the trim flap system was established and compared with the FE model fixed on the frame as mentioned in Section 3.2. The backshell model was taken from the FE model of the whole model of the Tianwen-1 Mars probe and corrected by static and modal experiments. It mainly consisted of the skin, bracket, and four frame beams, as shown in Figure 14, and was modeled by classical laminated plate theory with quadrilateral reduced integration elements of 100,759. Then, the FE model of the trim flap system built in Section 3.2 was imported into the backshell model and attached to the backshell through connector elements at the bracket and supports. The connector elements were used to simulate bolt connections, whose constraint reaction forces represented the impact force on the backshell caused by the deployment of the trim flap system.

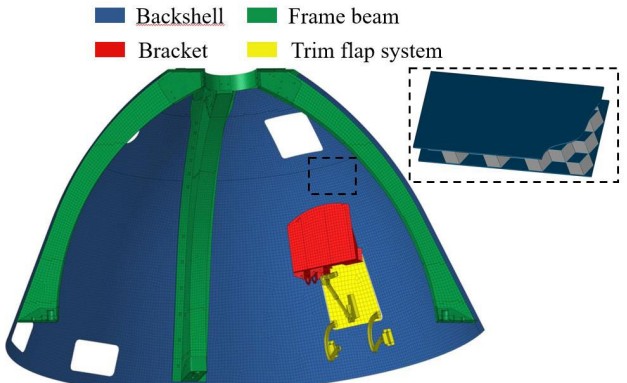

**Figure 14.** FE model of the trim flap system installed on the backshell.

The backshell was fabricated in the sandwich structure with a honeycomb core and Epoxy resin/M55 facets as highlighted in Figure 14, where layers 1 to 4 and 6 to 9 were composite material with the thickness of 0.08 mm per layer and every facet lay-up was

$[0°/45°/90°/−45°]_4$. The aluminum honeycomb core was simplified as a homogenous material with the thickness of 9.36 mm. Their mechanical properties are presented in Table 1. The bottom of the frame beam was fixed and other conditions were the same as the experimental setting.

Figure 15 shows the deployment process of the trim flap system mounted to the backshell. It was observed from Figure 15 that the flap panel could be successfully deployed, similar to the experimental and numerical results in Figure 8, however, a significant deformation occurred in the backshell skin near the bracket and supports (see the results from 0.09 s to 0.45 s). Figure 16a,b shows the comparisons between the FE model fixed on the frame and the one mounted to the backshell. As seen from Figure 16, although the response curves obtained from these two FE models showed a similar trend, large differences could still be found, especially in the peak values of the angular velocity and constraint reaction force of the latter, which were significantly higher than those of the former. The reason could be attributed to complex interactions between the backshell and the four-bar mechanism: (1) on one hand, due to the backshell flexibility, the impact force caused by the deployment of the four-bar mechanism induced the structure deformation at the local area near the bracket and supports, which changed the position of the four-bar mechanism, thus leading to an increase in deployment velocity; (2) on the other hand, the increase of the deployment velocity in turn resulted in a higher impact force, inducing a larger deformation. It could be inferred that the effect of backshell flexibility might be easily improved by increasing the local stiffness of the supports, bracket, and the nearby backshell skin to maintain the position of the four-bar mechanism, but this would inevitably cause a significant increase in weight. However, considering that the peak reaction force of 1226 N and *Dt* still met the design requirements, the above analysis results gave us the confidence that the current design could be acceptable.

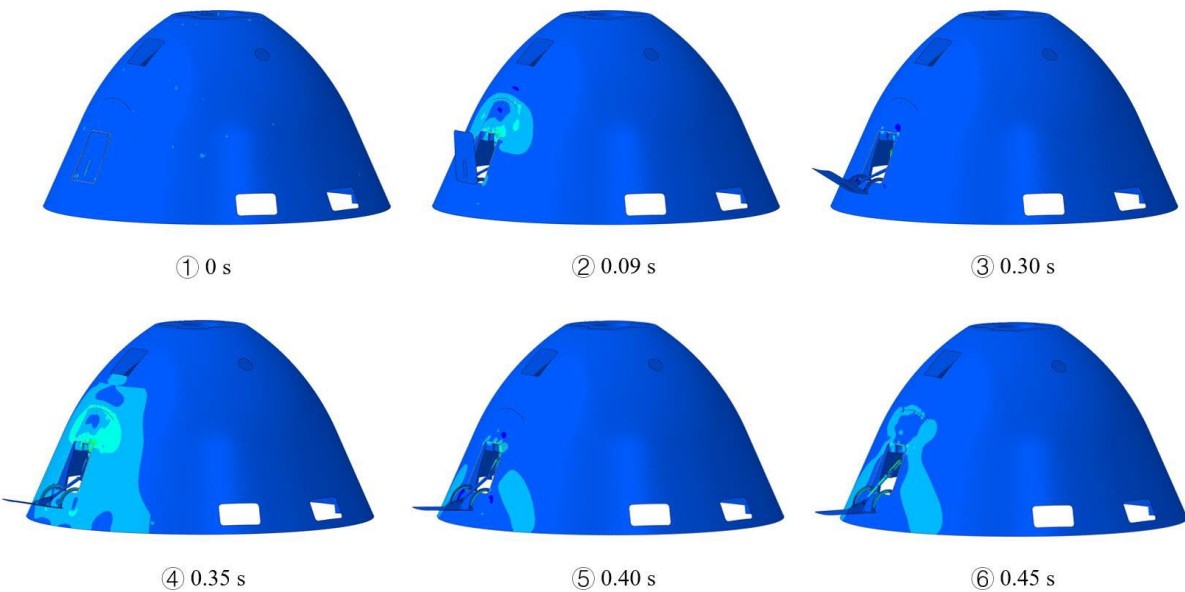

① 0 s  ② 0.09 s  ③ 0.30 s

④ 0.35 s  ⑤ 0.40 s  ⑥ 0.45 s

**Figure 15.** Deployment process of the trim flap system installed on the backshell.

Finally, this manufactured prototype was adopted and installed on the Tianwen-1 Mars probe, as shown in Figure 17a. The stowed and fully deployed status of the trim flap system just before the EDL phase and during the EDL phase are presented in Figure 17b,c, respectively, which were captured by the camera on the backshell. It could be observed that the trim flap system was successfully deployed during the EDL phase, helping the Tianwen-1 Mars probe to land successfully. The successful landing conclusively proved the effectiveness of the designed prototype and analysis methods.

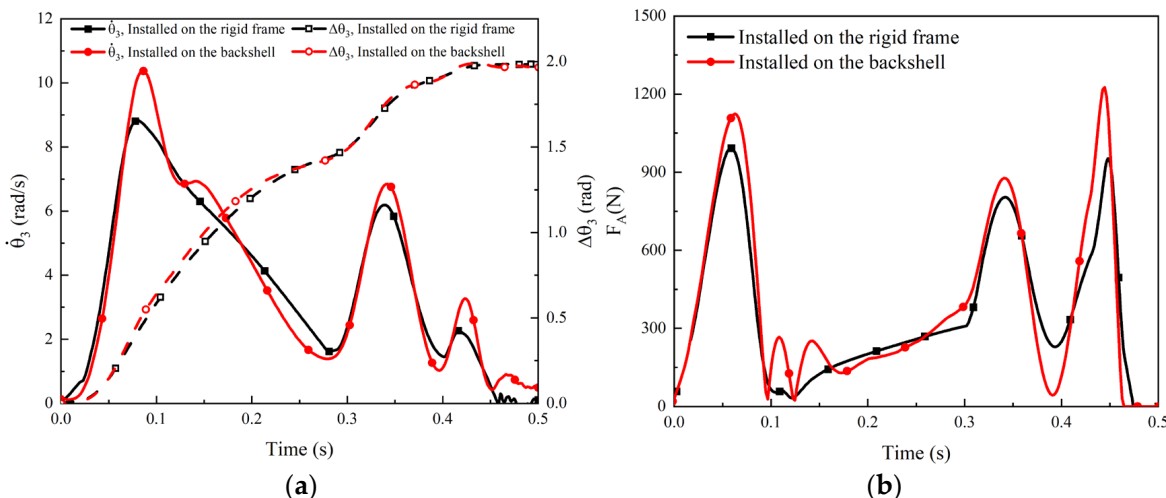

**Figure 16.** Comparison of the deployment performance between the FE model installed on the frame and backshell: (**a**) Deployment angle and angular velocity versus time of the flap panel; (**b**) Time history curve of the constraint reaction force.

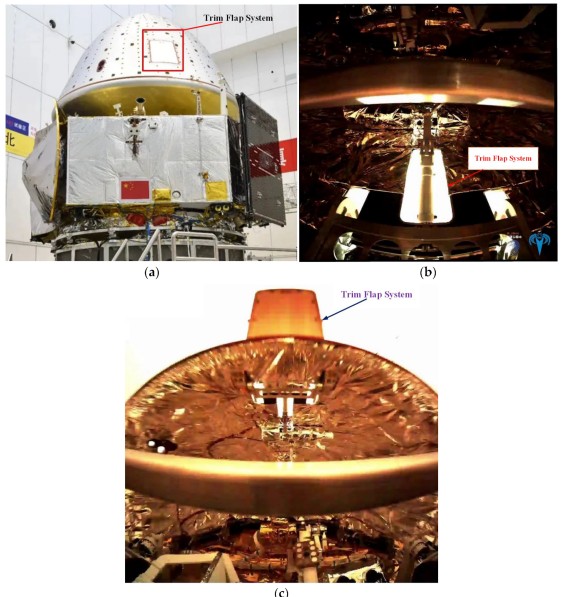

**Figure 17.** The trim flap system mounted on Tianwen-1 Mars probe: (**a**) Stowed configuration prior to launch; (**b**) Stowed configuration before the EDL phase; (**c**) Fully deployed configuration during the EDL phase.

## 5. Conclusions

In the present article, driven by engineering demand to satisfy the aerodynamic trim performance requirement of China's first Mars probe, Tianwen-1, in order to achieve precision landing, access landing sites at higher surface elevations, and increase payload mass, a deployable trim flap system was proposed and designed based on a single crank–rocker mechanism, which was stowed in the backshell during launch, exo-atmospherically deployed during the EDL phase, and remained in its deployed status during entry. According to the design, analysis, and deployment experiments of the physical prototype, the main conclusions were drawn as follows:

1. The deployable configuration was adopted for the design of the trim flap system, considering the limitation of the internal space of the Tianwen-1 Mars probe. The proposed trim flap system mainly consisted of the driving mechanism, crank, connecting

rod, curved beams, and flap panel, thus forming a single crank–rocker mechanism actuated by the driving mechanism to enable the flap panel deployment within 1.5 s and withstand entry aerodynamic load. The total mass of 10 kg was only about one-quarter of that of the MSL-I designed by NASA and less than one-thirtieth of the ejected ballast mass of MSL, which used the ballast mass CG offset method, allowing the Tianwen-1 Mars probe to carry more payload;

2. A theoretical model for the kinematic and dynamic analysis, as well as an FE model were established to evaluate and predict the deployment performance, as well as to provide guidance for a detailed structure design of the trim flap system;

3. A full-scale physical prototype of the proposed trim flap system was manufactured based on theoretical and FE analysis. The deployment experiments were conducted and the experimental results validated the effectiveness of the proposed trim flap system. Moreover, by comparing the results achieved from the experiment, theory and FE simulation, analysis results matched well with the experimental result qualitatively and quantitatively, demonstrating the validity of the proposed theoretical and FE models;

4. After being validated by the deployment experiment, the developed theoretical and FE models were implemented to investigate the effect of the nonlinear behavior of damping, acceleration environment, and backshell flexibility on the deployment performance of the trim flap system. It was found that the backshell flexibility could result in higher impact loads on the backshell due to the coupling of the backshell structure deformation and the motion of the four-bar mechanism, which was disadvantageous for the deployment performance. Additionally, the other two factors had insignificant effects on the deployment performance and thus could be ignored.

The current work builds a solid basis for utilizing the trim flap for Tianwen-1 and other Mars entry vehicles. The deployable trim flap system provided good aerodynamic trim performance to achieve a successful landing, making China the first country in the world to adopt the trim flap for Mars EDL. The proposed structure configuration and analysis methods could provide a reference for future human Mars missions and other solar system destinations employing the trim flap technology, in order to improve EDL performance.

**Author Contributions:** Conceptualization, X.L., W.R., Q.Y. and G.W.; methodology, Y.L. and D.H.; software, Y.L.; validation, Y.L., D.H. and Y.Q.; formal analysis, Y.L. and D.H.; investigation, X.L., Y.L. and D.H.; resources, X.L. and D.H.; data curation, Q.Y. and G.W.; writing—original draft preparation, Y.L. and D.H.; writing—review and editing, Y.L., D.H. and W.R.; visualization, Y.L. and D.H.; supervision, D.H. and W.R.; project administration, D.H. and Y.Q.; funding acquisition, X.L. and D.H. All authors have read and agreed to the published version of the manuscript.

**Funding:** This work is supported by the National Natural Science Foundation of China (No. 11872100, 52075020 and 51735009), National Science and Technology Major project, and Defense Industrial Technology Development Program (No. JCKY2018601B106 and JCKY2017205B032).

**Institutional Review Board Statement:** Not applicable.

**Informed Consent Statement:** Not applicable.

**Data Availability Statement:** Not applicable.

**Conflicts of Interest:** The authors declare no conflict of interest.

## Appendix A

The kinematic and dynamic analysis for the four-bar mechanism were as follows:

The dynamic equations of motion could be established using the Lagrange method. According to the geometry relationship as shown in Figure 3f, the geometric equations could be described as follows:

$$\Phi_1 = l_1 \sin \theta_1 + l_2 \sin \theta_2 + l_3 \sin \theta_3 - Y \tag{A1}$$

$$\Phi_2 = l_1 \cos\theta_1 - l_2 \cos\theta_2 + l_3 \cos\theta_3 + X \tag{A2}$$

According to the Lagrange method, the kinematic energy for AB, BC, and CDE could be expressed as:

$$T_1 = \frac{1}{2} J_1 \dot{\theta}_1^2 \tag{A3}$$

$$T_2 = \frac{1}{2} J_2 \dot{\theta}_2^2 + \frac{1}{2} m_2 [\alpha_1^2 \dot{\theta}_1^2 l_1^2 + \alpha_3^2 \dot{\theta}_3^2 l_3^2 - 2\alpha_1 \alpha_3 \dot{\theta}_1 \dot{\theta}_3 l_1 l_3 \cos(\theta_1 - \theta_3)] \tag{A4}$$

where $\alpha_1 = \frac{l_2 - r_2}{l_2}$, $\alpha_3 = \frac{r_2}{l_2}$

$$T_3 = \frac{1}{2} J_3 \dot{\theta}_3^2 \tag{A5}$$

where $m_1$, $m_2$, and $m_3$ represent the mass of AB, BC, and CDE, respectively. $J_1$, $J_2$, and $J_3$ represent the rotational inertias of AB around point A, BC around point B, and CDE around point D, respectively. $T_1$, $T_2$, and $T_3$ represent the kinetic energy of AB, BC and CDE, respectively.

According to the Lagrange method, the potential energy of the AB, BC, and CDE, respectively, are:

$$V_1 = m_1 g(Y - r_1 \sin\theta_1) \tag{A6}$$

$$V_2 = m_2 g[(l_2 - r_2)\sin\theta_2 + l_3 \sin\theta_3] \tag{A7}$$

$$V_3 = m_3 g r_3 \sin(\theta_3 - \gamma) \tag{A8}$$

where $V_1$, $V_2$, and $V_3$ represent the gravitational potential energy of AB, BC, and CDE, respectively. $r_1$, $r_2$, and $r_3$ represent the centroid position of AB, BC, and CDE, respectively. The value of $r_1$ is the distance between the centroid of AB and point A. The value of $r_2$ is the distance between the centroid of BC and point B. The value of $r_3$ is the distance between the centroid (point F) of CDE and point D. $\gamma$ is the angle of $\angle$FDC.

The total kinetic and potential energy of the mechanism is:

$$L = T_1 + T_2 + T_3 - V_1 - V_2 - V_3 \tag{A9}$$

The Lagrange equation for the mechanism could be defined as:

$$\frac{d}{dt}\left(\frac{\partial L}{\partial \dot{\theta}_i}\right) - \frac{\partial L}{\partial \theta_i} = Q_i + \lambda_1 \frac{\partial \Phi_1}{\partial \theta_i} + \lambda_2 \frac{\partial \Phi_2}{\partial \theta_i}, i = 1, 2, 3 \tag{A10}$$

where $Q_i$, ($i = 1, 2, 3$) is the generalized force: $Q_1$ is $M_A$, $Q_2$ is 0, and $Q_3$ is $M_D$, while $\lambda_1$ and $\lambda_2$ are Lagrangian multipliers. By substituting Equations (A1)–(A9) into Equation (A10) and then differentiating with respect to the generalized coordinates $\theta_i$, ($i = 1, 2, 3$), the kinematic equations of AB, BC and CDE could be obtained from Equations (A11)–(A13):

$$J_1 \ddot{\theta}_1 + \frac{1}{2} m_2 [2\alpha_1^2 \ddot{\theta}_1 l_1^2 - 2\alpha_1 \alpha_3 \ddot{\theta}_3 l_1 l_3 \cos(\theta_1 - \theta_3) + 2\alpha_1 \alpha_3 \dot{\theta}_3 l_1 l_3 \sin(\theta_1 - \theta_3)(\dot{\theta}_1 - \dot{\theta}_3)]$$
$$- m_2 \alpha_1 \alpha_3 \dot{\theta}_1 \dot{\theta}_3 l_1 l_3 \sin(\theta_1 - \theta_3) - m_1 g r_1 \cos\theta_1 = M_A + \lambda_1 l_1 \cos\theta_1 - \lambda_2 l_1 \sin\theta_1 \tag{A11}$$

$$J_2 \ddot{\theta}_2 + m_2 g(l_2 - r_2)\cos\theta_2 = \lambda_1 l_2 \cos\theta_2 + \lambda_2 l_2 \sin\theta_2 \tag{A12}$$

$$J_3 \ddot{\theta}_3 + \frac{1}{2} m_2 [2\alpha_3^2 \ddot{\theta}_3 l_3^2 - 2\alpha_1 \alpha_3 \ddot{\theta}_1 l_1 l_3 \cos(\theta_1 - \theta_3) + 2\alpha_1 \alpha_3 \dot{\theta}_1 l_1 l_3 \sin(\theta_1 - \theta_3)(\dot{\theta}_1 - \dot{\theta}_3)]$$
$$+ m_2 \alpha_1 \alpha_3 \dot{\theta}_1 \dot{\theta}_3 l_1 l_3 \sin(\theta_1 - \theta_3) + m_3 g r_3 \cos(\theta_3 + \gamma) + m_2 g l_3 \cos\theta_3$$
$$= M_D + \lambda_1 l_3 \cos\theta_3 - \lambda_2 l_3 \sin\theta_3 \tag{A13}$$

By combing Equations (A11) and (A12), Lagrangian multipliers $\lambda_1$ and $\lambda_2$ could be solved:

$$\lambda_1 = \frac{C_1 l_2 \sin\theta_2 + C_2 l_1 \sin\theta_1}{l_1 l_2 \sin(\theta_1 + \theta_2)} \tag{A14}$$

$$\lambda_2 = \frac{C_2 l_1 \cos\theta_1 - C_1 l_2 \cos\theta_2}{l_1 l_2 \sin(\theta_1 + \theta_2)} \tag{A15}$$

where $C_1$ and $C_2$ are:

$$C_1 = J_1\ddot{\theta}_1 + \frac{1}{2}m_2[2\alpha_1{}^2\ddot{\theta}_1 l_1{}^2 - 2\alpha_1\alpha_3\ddot{\theta}_3 l_1 l_3 \cos(\theta_1 - \theta_3) + 2\alpha_1\alpha_3\dot{\theta}_3 l_1 l_3 \sin(\theta_1 - \theta_3)(\dot{\theta}_1 - \dot{\theta}_3)]$$
$$- m_2\alpha_1\alpha_3\dot{\theta}_1\dot{\theta}_3 l_1 l_3 \sin(\theta_1 - \theta_3) - m_1 g r_1 \cos\theta_1 - M_A \tag{A16}$$

$$C_2 = J_2\ddot{\theta}_2 + m_2 g(l_2 - r_2)\cos\theta_2 \tag{A17}$$

After substituting Equations (A14)–(A17) into Equation (A13), Equation (A13) could be rewritten as:

$$J_3\ddot{\theta}_3 + \frac{1}{2}m_2[2\alpha_3{}^2\ddot{\theta}_3 l_3{}^2 - 2\alpha_1\alpha_3\ddot{\theta}_1 l_1 l_3 \cos(\theta_1 - \theta_3) + 2\alpha_1\alpha_3\dot{\theta}_1 l_1 l_3 \sin(\theta_1 - \theta_3)(\dot{\theta}_1 - \dot{\theta}_3)] +$$
$$m_2\alpha_1\alpha_3\dot{\theta}_1\dot{\theta}_3 l_1 l_3 \sin(\theta_1 - \theta_3) + m_3 g r_3 \cos(\theta_3 + \gamma) + m_2 g l_3 \cos\theta_3 = \tag{A18}$$
$$M_D + \frac{C_1 l_2 \sin\theta_2 + C_2 l_1 \sin\theta_1}{l_1 l_2 \sin(\theta_1 + \theta_2)} l_3 \cos\theta_3 - \frac{C_2 l_1 \cos\theta_1 - C_1 l_2 \cos\theta_2}{l_1 l_2 \sin(\theta_1 + \theta_2)} l_3 \sin\theta_3$$

The solutions for angle, angular velocity and angular acceleration could be obtained through the classic fourth order Runge–Kutta method by substituting Equations (4)–(6) and 24 into MATLAB Ode15i function.

Next, based on the force analysis in Figure 3f, the constraint reaction forces of each point could be obtained through equilibrium equation of forces and moments, as follows:

For AB,

$$F_{Ax} + F_{Bx} = m_1 a_{1x} \tag{A19}$$

$$F_{Ay} - m_1 g + F_{By} = m_1 a_{1y} \tag{A20}$$

$$m_1 g r_1 \cos\theta_1 + M_A - F_{By} l_1 \cos\theta_1 - F_{Bx} l_1 \sin\theta_1 = J_1\ddot{\theta}_1 \tag{A21}$$

For BC,

$$F_{Bx} + F_{Cx} = m_2 a_{2x} \tag{A22}$$

$$F_{By} - m_2 g + F_{Cy} = m_2 a_{2y} \tag{A23}$$

$$F_{Bx} r_2 \sin\theta_2 - F_{By} r_2 \cos\theta_2 - F_{Cx}(l_2 - r_2)\sin\theta_2 + F_{Cy}(l_2 - r_2)\cos\theta_2 = J_2\ddot{\theta}_2 \tag{A24}$$

For CDE,

$$F_{Cx} + F_{Dx} = m_3 a_{3x} \tag{A25}$$

$$F_{Cy} - m_3 g + F_{Dy} = m_3 a_{3y} \tag{A26}$$

$$F_{Cy} l_3 \cos\theta_3 + F_{Cx} l_3 \sin\theta_3 - m_3 g r_3 \cos(\theta_3 - \gamma) + M_D = J_3\ddot{\theta}_3 \tag{A27}$$

where $a_{1x}$ and $a_{1y}$ are the centroid acceleration of AB along the X and Y-directions, respectively; $a_{2x}$ and $a_{2y}$ are the centroid acceleration of BC along the X and Y-directions, respectively; $a_{3x}$ and $a_{3y}$ are the centroid acceleration of CDE along the X and Y-directions, respectively. They could be calculated from Equations (A28)–(A33):

For AB,

$$a_{1x} = -r_1\ddot{\theta}_1 \sin\theta_1 - r_1\dot{\theta}_1{}^2 \cos\theta_1 \tag{A28}$$

$$a_{1y} = -r_1\ddot{\theta}_1 \cos\theta_1 + r_1\dot{\theta}_1{}^2 \sin\theta_1 \tag{A29}$$

For BC,

$$a_{2x} = r_2\ddot{\theta}_2 \sin\theta_2 + r_2\dot{\theta}_2{}^2 \sin\theta_2 - l_1\dot{\theta}_1{}^2 \cos\theta_1 - l_1\ddot{\theta}_1 \sin\theta_1 \tag{A30}$$

$$a_{2y} = -r_2\ddot{\theta}_2 \cos\theta_2 + r_2\dot{\theta}_2{}^2 \sin\theta_2 + l_1\dot{\theta}_1{}^2 \sin\theta_1 - l_1\ddot{\theta}_1 \cos\theta_1 \tag{A31}$$

For CDE,

$$a_{3x} = r_3 \ddot{\theta_3} \sin(\theta_3 + \sigma) + r_3 \dot{\theta_3}^2 \cos(\theta_3 - \gamma) \tag{A32}$$

$$a_{3y} = r_3 \ddot{\theta_3} \cos(\theta_3 + \sigma) - r_3 \dot{\theta_3}^2 \sin(\theta_3 - \gamma) \tag{A33}$$

By solving the Equations (A19)–(A33), constraint reaction forces at each point could be determined.

The geometric parameters and mass properties of AB, BC, and CDE used for the kinematic and dynamic analysis are given in Table A1. The distances between point A and D along the *X*-direction and *Y*-direction were $X = 269.4$ mm and $Y = 440.5$ mm, respectively, while the initial value of $\theta_1$, $\theta_2$, and $\theta_3$ was 0.3688, 0.5846 and 2.2003, respectively. The feasibility of applying these design parameters were also validated by SolidWorks motion simulation to avoid geometric interferences with the backshell and other components.

**Table A1.** Geometric parameters for kinematic and dynamic analysis.

| | Length mm | Centroid Position mm | Mass g | Rotational Inertia g·mm$^2$ |
|---|---|---|---|---|
| AB | 190 | 109 | 294 | $4.86 \times 10^6$ |
| BC | 410 | 167 | 504 | $2.44 \times 10^7$ |
| CDE | 180 | 209 | 2569 | $2.07 \times 10^8$ |

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
