# Peer review of "Trim Flap System Design for Improving Ballistic-Lifting Entry Performance of the Tianwen-1 Mars Probe"

_aerospace, doi:10.3390/aerospace9060287_

Round 1

Reviewer 1 Report

There are still no enough publications addressing information and knowledge on Trim Flap Systems applied to Mars landing vehicles. This work presents a deployable trim flap system to meet the aerodynamical trim requirement of Tianwen-1 Mars probe but it can be applied to other kind of space vehicles and other solar system destination.  Theoretical and numerical models, as well as the construction of a physical structure configuration, are presented to investigate and improve the EDL performance. The agreement between theoretical, numerical and experimental results is amazing. They are the strengths of the manuscript. Undoubtedly, this is a sound, timely paper that presents important contributions employing the trim flap technology to improve EDL performance. The research was carefully and adequately framed and is presented in a transparent and comprehensive way. The manuscript is very well addressed, structured and written. Therefore, I recommend its publication.

Reviewer 2 Report

The paper Trim Flap System Design for improving ballistic-lifting entry performance of Tianwen-1 Mars Probe is about the design of deployable trim flap system. There are some points before consider for publication:

1- Authors claims that this work making China the first country in the world to utilize the trim flap technology for Mars EDL. The current paper is just a design without QA, QC, testing in vacum and Mars atmosphere. How they prove those thing for just a design ?

2- Again in text there are some lines directly mention a country : "due to China's weak technical background". How the authors have such knowledge?

3- The critical points of the current design are joint and used mechanism to move the arms. for example the revolute joint was employed to link Point A. But in conclusion not the results and safety factor is reported for them.

4- The design did not consider failure scenarios such as wind in Mars as the load could be applied or any safety analysis.

5- Sensitivity analysis are needed as well.

6- More discussion and data needed. For example mechanical plots with more detail.

Round 2

Reviewer 2 Report

Now is publishable.